# Stochastic Density Functional Theory on Lane Formation in Electric-Field-Driven Ionic Mixtures: Flow-Kernel-Based Formulation

**DOI:** 10.3390/e24040500

**Published:** 2022-04-01

**Authors:** Hiroshi Frusawa

**Affiliations:** Laboratory of Statistical Physics, Kochi University of Technology, Tosa-Yamada, Kochi 782-8502, Japan; frusawa.hiroshi@kochi-tech.ac.jp

**Keywords:** electric-field-driven systems, driven colloidal suspensions, lane formation, stationary correlation functions, flow kernel, stochastic density functional theory, the Dean–Kawasaki equation

## Abstract

Simulation and experimental studies have demonstrated non-equilibrium ordering in driven colloidal suspensions: with increasing driving force, a uniform colloidal mixture transforms into a locally demixed state characterized by the lane formation or the emergence of strongly anisotropic stripe-like domains. Theoretically, we have found that a linear stability analysis of density dynamics can explain the non-equilibrium ordering by adding a non-trivial advection term. This advection arises from fluctuating flows due to non-Coulombic interactions associated with oppositely driven migrations. Recent studies based on the dynamical density functional theory (DFT) without multiplicative noise have introduced the flow kernel for providing a general description of the fluctuating velocity. Here, we assess and extend the above deterministic DFT by treating electric-field-driven binary ionic mixtures as the primitive model. First, we develop the stochastic DFT with multiplicative noise for the laning phenomena. The stochastic DFT considering the fluctuating flows allows us to determine correlation functions in a steady state. In particular, asymptotic analysis on the stationary charge-charge correlation function reveals that the above dispersion relation for linear stability analysis is equivalent to the pole equation for determining the oscillatory wavelength of charge–charge correlations. Next, the appearance of stripe-like domains is demonstrated not only by using the pole equation but also by performing the 2D inverse Fourier transform of the charge–charge correlation function without the premise of anisotropic homogeneity in the electric field direction.

## 1. Introduction

Many industrial processes involve the transport of colloidal particles under external fields. For example, particles are driven by stirring or shearing, whereas other typical driving forces arise from gravity and an external electric field. The response of particles to the driving forces is a subject of considerable practical interest. In particular, electric-field-driven particles, with which we are concerned in this paper, play significant roles in biological ion channels, micro/nanofluidic devices for environmental and biomedical applications, and electrolyte-immersed porous electrodes for electrochemical applications [1,2,3]. We consider binary mixtures of symmetric ions under external electric fields as the electric-field-driven particles, which include not only oppositely charged colloidal mixtures but also electrolytes and room-temperature ionic liquids. Recently, the binary ionic mixtures under external fields are increasingly attracting much attention due to their diverse applications not only in chemistry and biology [1] but also in renewable energy devices such as batteries, supercapacitors, and separation media [3].

The driven binary mixtures, in which two populations of particles are driven in opposite directions, undergo an out-of-equilibrium transition [4,5,6,7,8,9,10,11,12,13,14,15,16,17,18]. For example, colloidal particles driven by a strong external field self-organize into strongly anisotropic stripe-like or layered structures (i.e., lanes), thereby representing a prototype of nonequilibrium phase transition [4,5,6,7,8,9,10,11,12,13,14,15,16,17,18,19]. The underlying mechanism of lane formation has been ascribed to the competition between thermodynamic tendency to mix binary colloids and kinetic preference to segregate colloids of the same kind for reducing collisions due to oppositely driven migrations [4,5,6,7,8,9,10]. Experimental and simulation studies have demonstrated that driven colloids provide a testbed for pattern formation occurring in many nonequilibrium systems; for the driven binary mixtures cover a variety of driven systems ranging from colloidal suspensions and electrolytes to active matter composed of autonomously moving agents such as pedestrians [4,5].

Experimental and simulation studies have also observed the band formation of like-charged colloidal particles, other than laning [11,12,13,14,15,16]. The bands are aligned in a direction non-parallel to the applied field direction and lead to a jammed state where the particles block each other’s motion. We have thus obtained dynamic phase diagrams of steady states of laned, jammed, and mixed structures formed by oppositely charged particles under a DC or AC electric field [5,6,7,8,9,10,11,12,13,14,15,16]. Especially under oscillatory electric field, laning generally occurs for a high enough field strength and a low oscillation frequency, whereas jammed and other non-laned structures emerge depending on the magnitude of the driving field and its oscillatory frequency [11,16]. The key observables to detect the emergence of such various structures in steady states are correlation functions; however, there are a few studies of electric-field-driven systems based on the correlation function analysis [20,21,22], and no attempts to address lane formation have been made. To perform the correlation function analysis, the stochastic density functional theory (DFT) [22,23,24,25,26,27,28,29,30,31,32,33] for lane formation needs to be developed.

Let us then provide a brief review on the dynamical density functional theory (DFT) developed to describe the overdamped dynamics of Brownian particles [23]. The dynamical DFT can treat a background flow by adding an advection term; however, an additional contribution needs to be included based on phenomenological arguments [9,10,17,18,19] for explaining the lane formation irrespective of whether the density functional equation used is deterministic or stochastic. The phenomenological term added to the dynamical density functional equation considers fluctuating motions around an external-field-driven migration of a particle when neglecting other particles [9,10,17,18,19]. There are two ways to add the fluctuating flow in an advection term. One method developed for sheared colloidal suspensions adds a particle-induced fluctuation flow to the velocity field [17,18,19]. A flow kernel [17,18,19,34,35,36,37,38] introduced in this approach allows us to treat non-local effects due to density fluctuations. The other treatment provides fluctuating currents transverse to the electric field for explaining the lane formation in oppositely charged colloids under external electric fields [9,10]. Both contributions capture the coupling between flow and interparticle interactions.

The above modifications for the description of lane formation belong to the deterministic DFT. An alternative approach to the dynamical DFT adopts the density functional equation with multiplicative noise, the so-called Dean–Kawasaki (DK) equation [23]. Recently, the stochastic DFT has found the usefulness of the DK equation linearized around a reference density [22,23,24,25,26,27,28,29,30,31,32,33]. The linearized DK equation allows us to compute correlation functions for density and charge fluctuations around uniform states. It is found from the correlation function analysis that density-density and charge-charge correlations are long-range correlated in the steady state [20,27,28]. The asymptotic decay of the stationary correlation functions exhibits a power-law behavior with a dipolar character, which gives rise to a long-range fluctuation-induced force acting on uncharged confining plates [27,28].

This paper aims to develop the stochastic DFT for explaining the lane formation of binary ionic mixtures. The electric-field-driven ionic mixtures are treated as the primitive model [1], implying that we consider non-equilibrium phenomena of either symmetric electrolytes or ionic liquids, rather than colloidal mixtures. The remainder of this paper is organized as follows. Section 2 provides the basic formalism for electric-field-driven binary ionic mixtures based on the deterministic and stochastic DFTs. In Section 3, we describe the purposes of reformulating and extending the previous formulation for lane formation. Furthermore, the remaining sections serve the two purposes. In Section 4, we revisit the linear stability analysis for lane formation based on the deterministic DFT [9,10,17,18], thereby unifying previous formulations of the above additional contributions due to fluctuating flows (the first purpose). Section 5 provides the stochastic DFT for lane formation. We obtain the Fourier transforms of correlation functions from the stochastic equations, thereby demonstrating that the charge-charge correlation function verifies the stability of lane structure both analytically and numerically (the second purpose). Section 6 presents a summary and conclusions.

## 2. Basic Formalism

### 2.1. Primitive Model

We consider a binary ionic mixture of cations and anions which have equal size and equal but opposite charge using the primitive model [1]. In this model, the *z*-valent ions in symmetric mixtures are modeled by equisized charged hard spheres of diameter σ immersed in a structureless and uniform dielectric medium with dielectric constant ϵ at a temperature *T*. The charged spheres interact via pairwise potential $v_{lm}\left(r\right)$ (l,m=1,2) where $v_{11}\left(r\right)$, $v_{12}\left(r\right)$, and $v_{22}\left(r\right)$ denote cation–cation, cation–anion, and anion–anion interaction potentials at a separation of r=|r|, respectively. Figure 1 presents a schematic of the 2D primitive model in Cartesian coordinates, illustrating that the electrophoretic force $zEk_{B}T$ is exerted on a single ion due to normalized electric field $E=Ee_{x}$ applied in the *x*-direction parallel to the unit vector $e_{x}$.

It is noted that this paper defines all of the energetic quantities, including the above interaction potentials, in units of $k_{B}T$. Correspondingly, the unit of zEσ, an energetic measure of electric field strength, is $k_{B}T$, and the interaction potential $v_{lm}\left(r\right)$ is represented by
(1)$$\begin{array}{c} v_{lm}\left(r\right)=\left\{\begin{array}{c} \infty \;(r<\sigma )\\ \left.{\left(-1\right)}^{l+m}z^{2}l_{B}\ln\left(\sigma /r\right)\;\left(r\geq \sigma \right)\right., \end{array}\right. \end{array}$$
using the Bjerrum length $l_{B}=e^{2}/\left(4\pi \epsilon k_{B}T\right)$, the length at which the bare Coulomb interaction between two monovalent ions is exactly $k_{B}T$.

The dynamical density functional theories focus on instantaneous concentrations $n_{l}\left(r,t\right)$ of cations (l=1) and anions (l=2) which vary depending on a time *t* as well as a position r. Here we also use the density vector N(r,t) defined by
(2)$$\begin{array}{cc} N(r,t) & =\left(\begin{array}{c} \rho (r,t)\\ q(r,t) \end{array}\right)=\left(\begin{array}{c} n_{1}\left(r,t\right)+n_{2}\left(r,t\right)\\ n_{1}\left(r,t\right)-n_{2}\left(r,t\right) \end{array}\right). \end{array}$$

While ρ(r,t) represents the number density of ions and is equal to $2\bar{n}$ in average, zeq(r,t) corresponds to the charge density whose average vanishes.

### 2.2. Stochastic Dft: Compact Matrix Forms

The stochastic DFT is based on the formulation that adds multiplicative noise term to the deterministic density functional Equations (Equation 45) previously used [22,23,24,25,26,27,28,29,30,31,32,33]. Therefore, the basic formalism of the stochastic DFT presented below inherits the formulation of the deterministic DFT given in Appendix A. In the stochastic DFT, the conservation equation reads
(3)$$\begin{array}{cc} & \partial_{t}n_{l}\left(r,t\right)+\nabla \cdot \left(v_{l}\left(r,t\right)n_{l}\left(r,t\right)\right)=-\nabla \cdot \left\{J_{l}^{\mu }\left(r,t\right)+J_{l}^{\zeta }\left(r,t\right)\right\}, \end{array}$$
where the deterministic current $J_{l}^{\mu }\left(r,t\right)$ is expressed by Equation (Equation 46) using the direct correlation function (DCF) $c_{lm}\left(r-r^{\prime }\right)$ between the *l*-th and *m*-th ions, and the stochastic density current $J_{l}^{\zeta }\left(r,t\right)$ is expressed as
(4)$$\begin{array}{c} J_{l}^{\zeta }\left(r,t\right)=-\sqrt{2Dn_{l}\left(r,t\right)}\zeta \left(r,t\right), \end{array}$$
using uncorrelated Gaussian noise fields ζ(r,t) characterized by
(5)$$\begin{array}{c} {\langle \zeta \left(r,t\right)\zeta {\left(r^{\prime },t^{\prime }\right)}^{T}\rangle }_{\zeta }=\delta \left(r-r^{\prime }\right)\delta \left(t-t^{\prime }\right), \end{array}$$
with the subscript “ζ” representing the Gaussian noise averaging in space and time.

We provide a compact matrix form of the stochastic equation with respect to N(r,t) through three steps as follows: (i) we obtain stochastic currents of ρ(r,t) and q(r,t) from Equations (Equation 46), ([Disp-formula FD5656-entropy-24-00500]) and (Equation 4), (ii) we write down a matrix form without external field, and (iii) we have the target equation of N-dynamics by adding the advection terms formulated from Equations (Equation 49) and ([Disp-formula FD56568-entropy-24-00500]).

First, we consider the relations (Equation 47) and (Equation 48) for the DCF, thereby transforming the sum of Equations (Equation 4) and (Equation 46) with Equation ([Disp-formula FD5656-entropy-24-00500]) to the linearized current as follows:(6)$$\begin{array}{cc} \left(\begin{array}{c} J_{\rho }\left(r,t\right)\\ J_{q}\left(r,t\right) \end{array}\right) & =\left(\begin{array}{c} J_{1}\left(r,t\right)+J_{2}\left(r,t\right)\\ J_{1}\left(r,t\right)-J_{2}\left(r,t\right) \end{array}\right) \end{array}$$
(7)$$\begin{array}{c} =-D\left(\begin{array}{c} \nabla \rho \left(r,t\right)-2\bar{n}\int d^{2}r^{\prime }\nabla c^{S}\left(r-r^{\prime }\right)\delta \rho \left(r,t\right)\\ \nabla q\left(r,t\right)-2\bar{n}\int d^{2}r^{\prime }\nabla c\left(r-r^{\prime }\right)q\left(r,t\right) \end{array}\right)-\sqrt{4D\bar{n}}\left(\begin{array}{c} \zeta (r,t)\\ \zeta^{\prime }\left(r,t\right) \end{array}\right), \end{array}$$
where $c^{S}\left(r\right)$ denotes the short-range part of the DCF (see also Equation (Equation 47)), $\delta \rho \left(r,t\right)=\nu_{1}\left(r,t\right)+\nu_{2}\left(r,t\right)$ with $\nu_{l}\left(r,t\right)=n_{l}\left(r,t\right)-\bar{n}$ (l=1 or 2), and $\zeta^{\prime }\left(r,t\right)$ satisfies the same statistics as the relation (Equation 5) for ζ(r,t).

Second, we rewrite Equation (Equation 3) into a compact matrix form,
(8)$$\begin{array}{cc} \partial_{t}N\left(k,t\right) & =-DK_{0}\left(k\right)N\left(-k,t\right)+\sqrt{4D\bar{n}}\;\eta \left(k\right), \end{array}$$
using
(9)$$\begin{array}{cc} \eta (r,t) & =\left(\begin{array}{c} \nabla \cdot \zeta (r,t)\\ \nabla \cdot \zeta^{\prime }\left(r,t\right) \end{array}\right), \end{array}$$
and
(10)$$\begin{array}{cc} K_{0}\left(k\right) & =\left(\begin{array}{cc} k^{2}\left\{1-2\bar{n}c^{S}\left(k,t\right)\right\} & 0\\ 0 & k^{2}\left\{1-2\bar{n}c\left(k,t\right)\right\} \end{array}\right)+O\left[\nu_{l}\right]. \end{array}$$

The matrix $K_{0}$ determines restoring forces in the absence of external field (E=0).

Third, we suppose that a fluctuating part ${\left(-1\right)}^{l-1}v^{\mathrm{fl}}\left(r,t\right)$ of $v_{l}\left(r,t\right)$ appears only in the *y*-direction, according to the previous treatments [9,10,17,18]. It follows from Equation (Equation 49) that
(11)$$\begin{array}{cc} \nabla \cdot n_{l}\left(r,t\right)v_{l}\left(r,t\right) & ={\left(-1\right)}^{l-1}\left\{DzE\partial_{x}n_{l}\left(r,t\right)+\bar{n}\partial_{y}v^{\mathrm{fl}}\left(r,t\right)\right\}+O\left[\nu_{l}\right], \end{array}$$
yielding
(12)$$\begin{array}{c} \left(\begin{array}{c} \nabla \cdot \left\{n_{1}\left(r,t\right)v_{1}\left(r,t\right)+n_{2}\left(r,t\right)v_{2}\left(r,t\right)\right\}\\ \nabla \cdot \left\{n_{1}\left(r,t\right)v_{1}\left(r,t\right)-n_{2}\left(r,t\right)v_{2}\left(r,t\right)\right\} \end{array}\right)=\left(\begin{array}{c} DzE\partial_{x}q\left(r,t\right)\\ DzE\partial_{x}\rho \left(r,t\right)+2\bar{n}\partial_{y}v^{\mathrm{fl}}\left(r,t\right) \end{array}\right). \end{array}$$

It is also noted that the Fourier transform of $v^{\mathrm{fl}}\left(r,t\right)$ reads
(13)$$\begin{array}{cc} v^{\mathrm{fl}}\left(k,t\right) & =G_{y}\left(k\right)q\left(-k,t\right),\\ G_{y}\left(k\right) & =\int d^{2}r\;G_{y}\left(r\right)e^{-ik\cdot r}\\ & =-i\int d^{2}r\;G_{y}\left(r\right)\sin\left(k\cdot r\right) \end{array}$$
(14)$$\begin{array}{c} =-ia(k), \end{array}$$
where we have used in the second line that the flow kernel $G_{y}\left(r\right)$ is an odd function satisfying $\int d^{2}rG_{y}\left(r\right)=0$
where we have used in the second line that the flow kernel $G_{y}\left(r\right)$ is an odd function satisfying $\int d^{2}rG_{y}\left(r\right)=0$ [17,18]. Combining Equations (Equation 8) to (Equation 14), we find the advected form of stochastic equation for N(r,t) under external electric field as follows:(15)$$\begin{array}{cc} \partial_{t}N\left(k,t\right) & =-DK\left(k\right)N\left(-k,t\right)+\sqrt{4D\bar{n}}\;\eta \left(k\right), \end{array}$$
where the matrix K(k) is given by
(16)$$\begin{array}{cc} K(k) & =K_{0}\left(k\right)+K_{v}\left(k\right), \end{array}$$
adding the advection matrix,
(17)$$\begin{array}{cc} K_{v}\left(k\right) & =\left(\begin{array}{cc} 0 & ik_{x}zE\\ ik_{x}zE & -2D\tilde{E}k_{y}^{2} \end{array}\right), \end{array}$$
with a new parameter $a\left(k\right)=-Dk_{y}\tilde{E}$ being introduced (see Equations (Equation 53)–(Equation 57) for details). Equation (Equation 13) including the above advection term is expected to form the basis of the stochastic DFT that is capable of addressing the lane formation.

## 3. Our Aim

In what follows, we assess and extend the deterministic DFT on lane formation in binary ionic mixtures with external electric fields applied. There are three reasons for revisiting the deterministic approaches. First, it is necessary to clarify the consistency between previous formulations. We have introduced the flow kernel in Equation ([Disp-formula FD56568-entropy-24-00500]) to describe a fluctuating velocity; however, this treatment has applied to one-component systems, sheared colloidal suspensions [17,18]. Therefore, the flow-kernel-based formulation for electric-field-driven mixtures needs to be developed. Second, the linear stability analysis [9,10,17,18] based on the dispersion relation (see Appendix B) has supposed that wavenumber is a real value, though wavenumber is a complex value in general [39,40,41,42]. It remains to be investigated to make the linear stability analysis while considering the imaginary part of wavenumber that determines the decay length of spatial modulation. The last reason is that the stochastic DFT, an extension of the deterministic DFT, makes it possible to investigate the mechanism of lane formation from density-density and charge-charge correlations.

Thus, we aim to obtain the correlation functions from the stochastic DFT for achieving the following two purposes.

*(i) Relationship between the deterministic and stochastic DFTs*—The first purpose is to understand the above dispersion relations in terms of the charge-charge correlation function. We will reveal the connection between those used in the deterministic DFT and the pole equation to find the oscillatory wavelength of charge-charge correlations.

*(ii) On the uniformity of lanes in terms of correlation function analysis based on the stochastic DFT*—The second purpose is to validate the approximation necessary to explain lane formation using the linear stability analysis [9,10,17,18]. Previous studies on lane phenomena have neglected the charge density modulation of the *x*-direction. In other words, $k_{x}=0$ has been assumed when investigating the charge density modulation in the transverse direction to the electric field. The correlation function analysis allows for the investigation of oscillatory decay behaviors such as the oscillatory wavelength ($\lambda_{x}^{*}$) along the electric field. Figure 2 is a schematic of this, which illustrates the emergence of the charge modulation, or the oscillatory charge-charge correlation due to the lane formation. We perform both the asymptotic analysis of charge–charge correlations for point charges (i.e., the primitive model at σ=0) and the 2D inverse Fourier transform of the stationary charge–charge correlation function for charged hard spheres (i.e., the primitive model for σ≠0). While the asymptotic analysis will prove that $\lambda_{x}^{*}$ diverges at the stabilization condition of lane structure, the real-space representation of the charge–charge correlation function will clarify the decay of oscillatory correlations in lane structures as a result of the inverse Fourier transform.

## 4. Correlation Functions Determined by the Stochastic Dft

### 4.1. Stationary Condition of Correlation Functions

The stochastic formulation allows us to provide the Fourier transforms of correlation functions for ρ(r,t) and q(r,t) at equal times [20,22,26,27,28,29]. These correlation functions are defined using N(k,t) as
(18)$$\begin{array}{cc} C(k,t) & ={\langle N\left(k,t\right)N{\left(-k,t\right)}^{T}\rangle }_{\zeta }\\ & =\left(\begin{array}{cc} {\langle \rho (k)\rho (-k,t)\rangle }_{\zeta } & {\langle q(k)\rho (-k,t)\rangle }_{\zeta }\\ {\langle \rho (k)q(-k,t)\rangle }_{\zeta } & {\langle q(k)q(-k,t)\rangle }_{\zeta } \end{array}\right)\\ & =\left(\begin{array}{cc} C_{\rho \rho }\left(k,t\right) & C_{q\rho }\left(k,t\right)\\ C_{\rho q}\left(k,t\right) & C_{qq}\left(k,t\right) \end{array}\right). \end{array}$$

The compact form (Equation 7) of the stochastic equation for N(k,t) is solved to obtain [20,25,26,27,28,29]
(19)$$\begin{array}{c} N\left(k,t\right)=\left\{\int_{-\infty }^{t}ds\;e^{-DK(k)(t-s)}\right\}\sqrt{4D\bar{n}}\;\eta \left(k\right), \end{array}$$
where Equations (Equation 5) and (Equation 9) provide
(20)$$\begin{array}{c} \langle \eta \left(k,t\right)\eta {\left(-k,t\right)}^{T}\rangle ={\left(2\pi \right)}^{2}\left(\begin{array}{cc} k^{2}\delta \left(t-t^{\prime }\right) & 0\\ 0 & k^{2}\delta \left(t-t^{\prime }\right) \end{array}\right). \end{array}$$

Plugging Equations (Equation 17) and (Equation 18) into the definition (Equation 16), we have
(21)$$\begin{array}{c} C\left(k,t\right)=\overset{t}{\underset{-\infty }{\;\int \int \;}}ds\;ds^{\prime }\;e^{-DK(t-s)}DR\;e^{-DK^{†}\left(t-s^{\prime }\right)}, \end{array}$$
where it follows from the relation (Equation 18) that
(22)$$\begin{array}{c} R\left(k\right)={\left(2\pi \right)}^{2}\left(\begin{array}{cc} 4\bar{n}k^{2} & 0\\ 0 & 4\bar{n}k^{2} \end{array}\right). \end{array}$$

It has been shown that the stationary condition dC(k,t)/dt=0 for the expression (Equation 19) reads [20,22,26,27,28,29]
(23)$$\begin{array}{c} KC+CK^{†}=R. \end{array}$$

The four matrix elements of C, or the four kinds of correlation functions in Equation (Equation 16), can be determined by four simultaneous equations generated from the above stationary condition (Equation 21) (see Appendix C for details).

### 4.2. Obtained Forms of Stationary Correlation Functions

As derived in Appendix C, Equation (Equation 21) yields the density–density and charge–charge correlation functions at equal times, $C_{\rho \rho }^{\mathrm{st}}\left(k\right)$ and $C_{qq}^{\mathrm{st}}\left(k\right)$, as follows:(24)$$\begin{array}{cc} \frac{1}{{\left(2\pi \right)}^{2}}\left(\begin{array}{c} C_{\rho \rho }^{\mathrm{st}}\left(k\right)\\ C_{qq}^{\mathrm{st}}\left(k\right) \end{array}\right) & =\frac{2\bar{n}k^{2}}{\left(\alpha +\beta \right)\left(\alpha \beta +\gamma^{2}\right)}\;\left(\begin{array}{cc} \beta \left(\alpha +\beta \right)+\gamma^{2} & \gamma^{2}\\ \gamma^{2} & \alpha \left(\alpha +\beta \right)+\gamma^{2} \end{array}\right)\left(\begin{array}{c} 1\\ 1 \end{array}\right), \end{array}$$
where
(25)$$\begin{array}{cc} \alpha & =k^{2}\left\{1-2\bar{n}c^{S}\left(k,t\right)\right\},\\ \beta & =-2\tilde{E}k_{y}^{2}+\beta_{0},\\ \gamma & =k_{x}zE, \end{array}$$
using $\beta_{0}=k^{2}/S\left(k\right)=k^{2}\left\{1-2\bar{n}c\left(k,t\right)\right\}$. In what follows, two limiting cases are considered for $C_{\rho \rho }^{\mathrm{st}}\left(k\right)$ and $C_{qq}^{\mathrm{st}}\left(k\right)$: (i) we confirm that these converge to the equilibrium correlation functions of electrolytes at E=0 and σ=0, and (ii) we see the dispersion relation given by Equation (Equation 59) in terms of the stationary correlation functions at $k_{x}=0$, according to the approximation (Equation 51).

Before proceeding, we need to connect the long-range part $c^{L}\left(r-r^{\prime }\right)$ of the DCF and the Coulomb potential ψ(r to provide the Poisson-like equation and the Debye–Hückel screening length. In general, $c^{L}\left(k\right)$ is expressed as
(26)$$\begin{array}{c} -c^{L}\left(k\right)=\frac{4\pi z^{2}l_{B}}{k^{2}}\omega \left(-k\right), \end{array}$$
using the weight function ω(k). For instance, ω(k)=cos(kσ) with k=|k| is a well-known form of the 3D primitive model [26,39,40]. Thus, the Poisson equation is generalized to the finite-spread type [43,44]:(27)$$\begin{array}{c} \nabla^{2}\psi \left(r,t\right)=-4\pi z^{2}l_{B}\int d^{2}r^{\prime }\omega \left(r-r^{\prime }\right)\;q\left(r^{\prime },t\right), \end{array}$$
when defining the Coulomb potential ψ(r,t) as
(28)$$\begin{array}{c} \psi \left(r,t\right)=-\int d^{2}r^{\prime }c^{L}\left(r-r^{\prime }\right)\;q\left(r^{\prime },t\right). \end{array}$$

It follows that
(29)$$\begin{array}{c} k^{2}\left\{1-2\bar{n}c^{L}\left(k\right)\right\}q\left(-k,t\right)=\left\{k^{2}+{\bar{\kappa }}^{2}\omega \left(k\right)\right\}q\left(-k,t\right), \end{array}$$
where
(30)$$\begin{array}{c} {\bar{\kappa }}^{2}=8\pi l_{B}z^{2}\bar{n} \end{array}$$
with ${\bar{\kappa }}^{-1}$ denoting the conventional Debye–Hückel screening length.

First, we consider equilibrium electrolytes. Since we have γ=0 and $\tilde{E}=0$ at E=0, Equation (Equation 22) is reduced to
(31)$$\begin{array}{cc} \frac{1}{{\left(2\pi \right)}^{2}}\lim_{E\to 0}\left(\begin{array}{c} C_{\rho \rho }^{\mathrm{st}}\left(k\right)\\ C_{qq}^{\mathrm{st}}\left(k\right) \end{array}\right) & =2\bar{n}k^{2}\;\left(\begin{array}{c} 1/\alpha \\ 1/\beta_{0} \end{array}\right). \end{array}$$

We also have $c^{S}\left(k\right)=0$, ω(k)=1, and $\beta_{0}=k^{2}+{\bar{\kappa }}^{2}$ at σ=0. Hence Equation (Equation 29) reads
(32)$$\begin{array}{cc} \frac{1}{{\left(2\pi \right)}^{2}}\lim_{E,\;\sigma \to 0}\left(\begin{array}{c} C_{\rho \rho }^{\mathrm{st}}\left(k\right)\\ C_{qq}^{\mathrm{st}}\left(k\right) \end{array}\right) & =2\bar{n}\;\left(\begin{array}{c} 1\\ k^{2}/\left(k^{2}+{\bar{\kappa }}^{2}\right) \end{array}\right), \end{array}$$
thereby confirming that the charge–charge correlation function $\lim_{E,\;\sigma \to 0}C_{qq}^{\mathrm{st}}\left(r\right)$ satisfies not only the electroneutrality but also the Stillinger–Lovett second-moment condition [27,28,29].

## 5. Lane Formation in Terms of Charge–Charge Correlation Function

### 5.1. Asymptotic Behavior of Charge–Charge Correlations

The lane formation has been investigated for $k_{x}=0$, according to previous studies on lane formation. This implies that the density modulation along the electric field (or the *x*–direction) at a given *y*–coordinate is negligible. Equation (Equation 22) at $k_{x}=0$ transforms to
(33)$$\begin{array}{cc} \frac{1}{{\left(2\pi \right)}^{2}}\lim_{k_{x}\to 0}\left(\begin{array}{c} C_{\rho \rho }^{\mathrm{st}}\left(k\right)\\ C_{qq}^{\mathrm{st}}\left(k\right) \end{array}\right) & =2\bar{n}k_{y}^{2}\;\left(\begin{array}{c} 1/\alpha \\ 1/\beta \end{array}\right), \end{array}$$
with $k^{2}$ being replaced by $k_{y}^{2}$ in Equation (Equation 23). Focusing on the pole equation (i.e., β=0) for $C_{qq}^{\mathrm{st}}\left(k\right)$, we have
(34)$$\begin{array}{c} 0=-2\tilde{E}{\left(k_{y}^{*}\sigma \right)}^{2}+{\left(k_{y}^{*}\sigma \right)}^{2}\left\{1-2\bar{n}c\left(k_{y}^{*},t\right)\right\}. \end{array}$$

It is noted that the above Equation (Equation 32) is identified with the key equation previously used for determining the mean wavelength $\lambda_{y}^{*}$ of lanes, which has been referred to as the dispersion relation (Equation 56) at $\tilde{\omega }=0$ in terms of the linear stability analysis [9,10,17,18] (see Appendix B).

The advantage over the linear stability analysis is that the pole Equation (Equation 32) provides the long-range behavior of density profile, or the decay length and oscillatory wavelength in the asymptotic decay of charge–charge correlation function [39,40,41,42]. In particular, the correlation function analysis ensures the stability of steady-state lane structure only when a solution to Equation (Equation 32) has a purely real wavenumber $k_{y}^{*}$, which is the case with the above linear stability analysis.

Furthermore, the asymptotic decay analysis of charge–charge correlations allows us to validate the supposition of uniformity along the electric field (or $k_{x}=0$) in the anisotropic lane structure as given in Figure 2. To assess the validity of $k_{x}=0$, we evaluate the inverse Fourier transform of stationary charge–charge correlation function $C_{qq}^{\mathrm{st}}\left(k\right)$ as follows:(35)$$\begin{array}{c} \frac{1}{2\pi }C_{qq}^{\mathrm{st}}\left(x,k_{y}\right)=\frac{1}{2\pi }\left[\frac{1}{2\pi }\int dk_{x}\;e^{ik_{x}x}C_{qq}^{\mathrm{st}}\left(k\right)\right]. \end{array}$$

We evaluate this inverse Fourier transform using the approximate form of $C_{qq}^{\mathrm{st}}\left(k\right)$ for $k_{x}{\bar{\kappa }}^{-1}\ll 1$ and σ=0. The denominator given in Equation (Equation 22) is approximated by
(36)$$\begin{array}{c} \left(\alpha +\beta \right)\left(\alpha \beta +\gamma^{2}\right)=\left\{\left(2-2\tilde{E}\right)k_{y}^{2}+{\bar{\kappa }}^{2}\right\}\left\{\left(k_{x}^{2}+k_{y}^{2}\right)\left\{\left(1-2\tilde{E}\right)k_{y}^{2}+{\bar{\kappa }}^{2}\right\}+{\left(zEk_{x}\right)}^{2}\right\}. \end{array}$$

Therefore, the pole equation $\alpha \beta +\gamma^{2}=0$ for $k_{x}^{*}$ yields
(37)$$\begin{array}{c} k_{x}^{*}=k_{y}\sqrt{\frac{\left(2\tilde{E}-1\right)k_{y}^{2}{\bar{\kappa }}^{-2}-1}{z^{2}E^{2}{\bar{\kappa }}^{-2}-\left(2\tilde{E}-1\right)k_{y}^{2}{\bar{\kappa }}^{-2}+1}}, \end{array}$$
providing the wavelength $\lambda_{x}^{*}=2\pi /k_{x}^{*}$ when $k_{x}^{*}$ is a purely real value for $zE{\bar{\kappa }}^{-1}>1$, $\tilde{E}∼zE\sigma$ (see Equation (Equation 60)), and $k_{y}{\bar{\kappa }}^{-1}<1$.

The asymptotic analysis allows us to provide the long-range oscillatory behavior of $C_{qq}^{\mathrm{st}}\left(x,k_{y}\right)$ as follows:(38)$$\begin{array}{c} C_{qq}^{\mathrm{st}}\left(x,k_{y}\right)∼\cos\left(\frac{2\pi x}{\lambda_{x}^{*}\left(k_{y}\right)}+\delta \right). \end{array}$$

It should be noted that the pole Equation (Equation 32) for $k_{y}^{*}$ reads
(39)$$\begin{array}{c} 0=2\tilde{E}{\left(k_{y}^{*}\sigma \right)}^{2}-\left\{{\left(k_{y}^{*}\sigma \right)}^{2}+{\left(\bar{\kappa }\sigma \right)}^{2}\right\} \end{array}$$
in the present case. We obtain from Equation (Equation 37)
(40)$$\begin{array}{c} \lambda_{y}^{*}=\frac{2\pi }{k_{y}^{*}}=2\pi {\bar{\kappa }}^{-1}\sqrt{2\tilde{E}-1}. \end{array}$$

Combining Equations (Equation 35) and (Equation 37), we also find
(41)$$\begin{array}{c} \lim_{k_{y}\to k_{y}^{*}}k_{x}^{*}=0. \end{array}$$

Namely, we have
(42)$$\begin{array}{c} \lim_{k_{y}\to k_{y}^{*}}\lambda_{x}^{*}\left(k_{y}\right)\to \infty \end{array}$$
when forming the lane structure with its period of $\lambda_{y}^{*}=2\pi /k_{y}^{*}$. Thus, it is verified analytically that each lane is uniform along the electric field the present approximation (Equation 53) as far as point charges (σ=0) are considered.

The expression (Equation 38) of $\lambda_{y}^{*}$, or the lane width, reveals the underlying physics of lane formation. Each lane has the energetic cost of Coulomb repulsions due to clustering of either cations or anions, which explains why lanes can be wider as ${\bar{\kappa }}^{-1}$ is shorter and the screening of Coulomb interactions is stronger. Despite this energetic cost, the lane formation is favored because collisions due to oppositely driven migrations are reduced by segregation of cations or anions. The kinetic preference is enhanced by increasing the strength of external field; accordingly, Equation (Equation 38) implies that the lane width is larger with increase of $\tilde{E}$.

### 5.2. Charge–Charge Correlations on 2D Cross Section of the 3D Primitive Model

The preceding subsection has analytically demonstrated that the dispersion relation based on the conventional linear stability analysis [9,10,17,18] is equivalent to the asymptotic decay analysis of the charge–charge correlation function. We have also verified that the dispersion relation applies to the emergence of a lane structure for point charges. Turning our attention to charged hard spheres of finite size, however, it remains to be validated whether we can neglect the decay of charge–charge correlations. At least, for the 3D primitive model in the absence of an electric field, theoretical and simulation studies have found that oscillatory decay of charge-charge correlations has been observed beyond the Kirkwood crossover condition [39,40,41,42]. In terms of the asymptotic decay analysis, the solution to the pole equation becomes complex at the Kirkwood crossover when considering the wavenumber-dependence of ω(k), and the imaginary part of the solution represents the finite decay length of charge-charge correlations.

To investigated the oscillatory decay behavior in the presence of an external field, we examine the stationary charge–charge correlation function $C_{qq}^{\mathrm{st}}\left(k\right)$ concerning a 2D cross-section of the 3D primitive model. Figure 3 represents a schematic of the present 3D system. From Figure 3, we can see that the xy plane in Figure 1 corresponds to the cross-section formed by the *x*- and *y*-axes embedded in this 3D system. The advantage of considering the 3D primitive model is that we can use the analytical form of DCF: the long-range part is given by Equation (Equation 24) with
(43)$$\begin{array}{cc} \omega (k) & =\cos(k\sigma ), \end{array}$$
whereas the short-range part reads
(44)$$\begin{array}{cc} -c^{S}\left(k\right) & =-\frac{4\pi \sigma }{k^{2}}\left\{\cos\left(k\sigma \right)-\frac{\sin(k\sigma )}{k\sigma }\right\}, \end{array}$$
in the modified mean spherical approximation [45]. Upon introducing the 3D volume fraction $\phi =\pi \sigma^{3}\bar{n}/6$, Equations (Equation 24) and (Equation 42) transform Equation (Equation 23) to
(45)$$\begin{array}{c} \alpha \sigma^{2}={\left(k\sigma \right)}^{2}-48\phi \left\{\cos\left(k\sigma \right)-\frac{\sin(k\sigma )}{k\sigma }\right\} \end{array}$$
and
(46)$$\begin{array}{cc} \beta \sigma^{2} & =-2\tilde{E}{\left(k_{y}\sigma \right)}^{2}+{\left(k\sigma \right)}^{2}+{\left(\bar{\kappa }\sigma \right)}^{2}\omega \left(k\right)\\ & =-2\tilde{E}{\left(k_{y}\sigma \right)}^{2}+{\left(k\sigma \right)}^{2}+48\phi \left(\frac{z^{2}l_{B}}{\sigma }\right)\omega \left(k\right). \end{array}$$

Therefore, under the simplification of $z^{2}l_{B}/\sigma =1$, the expressions (Equation 22), (Equation 23), (Equation 43) and (Equation 44) for $C_{qq}^{\mathrm{st}}\left(k\right)$ imply that the inverse Fourier transform of $C_{qq}^{\mathrm{st}}\left(k\right)$ depend on the three control parameters: zEσ, $\tilde{E}$, and ϕ.

Let the 3D wavevector be $k=(k_{x},k_{y},k_{z})$ which has $k_{z}$–component in addition to $k_{x}$–and $k_{y}$–components. However, we set $k_{z}=0$, which leads to the consideration of charge-charge correlations averaged over the *z*–axis density distribution in Figure 3 [22]. Accordingly, we can perform the 2D inverse Fourier transform of $C_{qq}^{\mathrm{st}}\left(k\right)$ in the Cartesian coordinates similar to those given in Figure 1. Figure 4 shows some of the results. Figure 4a,b on the left side are the results considering the presence of $v^{\mathrm{fl}}\left(r,t\right)$ with $\tilde{E}=0.484$. Meanwhile, Figure 4c,d on the right side ignore the fluctuating part with $\tilde{E}=0$. The other parameters are common to the results on the left and right sides. Namely, zEσ=1 for all of the results in Figure 4, ϕ=0.05 and $\bar{\kappa }\sigma =1.55$ in Figure 4a,c, and the concentration is increased by 10% in Figure 4b,d: the screening effect of Coulomb interactions is enhanced to $\bar{\kappa }\sigma =1.62$ due to ϕ=0.055 in Figure 4b,d.

We can draw three conclusions from comparing the results in Figure 4.

First, Figure 4a,b verify the lane formation of binary ionic mixtures in terms of charge-charge correlations. Especially in Figure 4b, we observe no decay of correlations in the electric-field direction over the length scale of 10 times the diameter of charged hard spheres. The oscillatory charge-charge correlations demonstrate that each lane of the 3D primitive model can be homogeneous in the electric-field direction, which agrees with the analytical investigations in Section 5.1.

Second, comparison between Figure 4a,b suggests the underscreening behavior [39,40,41,42]. On the one hand, Figure 4a indicates that the stationary charge–charge correlation function converges to zero far from the origin of (0,0), thereby illustrating an oscillatory decay behavior. In Figure 4b, on the other hand, we observe little change in the heat map color along the electric field direction. In other words, the purely oscillatory behavior, which is the premise of the linear stability analysis previously made, is demonstrated in Figure 4b. This change from Figure 4a to Figure 4b suggests that the decay length is longer as the volume fraction ϕ, or the ion concentration, increases similarly to the underscreening behavior in binary ionic mixtures with no electric field applied above the Kirkwood crossover where the equilibrium charge–charge correlation function exhibits oscillatory decays [39,40,41,42]. We have confirmed such underscreening behavior with an electric field applied.

Third, the difference between the results in Figure 4 on the left and right sides reveals that anisotropic oscillatory correlations, which reflect the lane formation, disappear in Figure 4c,d because of the absence of the fluctuating flow, $v^{\mathrm{fl}}\left(r,t\right)$, given by Equations (13) and (14). It is also important to note that the scale of the color bar on the right side is $10^{-2}$ times the scale on the left side. In other words, $C_{qq}^{\mathrm{st}}\left(x,y\right)$ is almost zero in Figure 4c,d. The weak charge–charge correlations imply that electric-field-driven binary ionic mixtures are uniform in the absence of the fluctuating flow which arises from collisions due to oppositely driven migrations of cations and anions.

## 6. Summary and Conclusions

The charge–charge correlation function studied so far can be detected using X-ray and/or neutron scattering experiments [46]. We would therefore like to evaluate experimental conditions that are consistent with the numerical results in Figure 4. For example, we consider $\left(z,\epsilon ,\sigma ,l_{B}\right)=\left(1,\;65,\;0.8\;\mathrm{nm},\;0.86\;\mathrm{nm}\right)$ as an room-temperature ionic diluted with propylene carbonate. It follows that $z^{2}l_{B}/\sigma \approx 1.1$ in correspondence with the supposition that $z^{2}l_{B}/\sigma =1$ in Figure 4. Also, the parameters, zEσ=1.0 and ϕ=0.05 (or $\bar{\kappa }\sigma =1.55$), used in Figure 4a read $E\approx 3.2\times 10^{7}$ V/m and 0.31 M, respectively, for the room-temperature ionic liquid. These are plausible values according to previous simulation and experimental studies [47,48]; in particular, it is interesting to note that molecular dynamics simulations of room-temperature ionic liquids have revealed that $E∼10^{7}$ V/m corresponds to a boundary value beyond which the ionic liquids are reorganized into nematic-like order and exhibit anisotropic dynamics [49].

Finally, we summarize the results presented so far, according to the two purposes mentioned in Section 3, the section of our aim.

*(i) Relationship between the deterministic and stochastic DFTs*—The wavenumber appearing in the dispersion relation (Equation 56) can actually be a complex number. It is appropriate for understanding the underlying physics of the complex wavenumber to see correlation functions instead of the dispersion relation. Hence, we have addressed the first purpose using the stochastic DFT for lane formation, thereby allowing us to obtain density–density and charge–charge correlation functions in a steady state. We have demonstrated that the asymptotic analysis of the charge–charge correlation function is equivalent to the linear stability analysis based on the dispersion relation (see also Appendix B). Specifically, the pole equation used in the asymptotic analysis proved equivalent to the lane stability condition obtained from the dispersion relation. The analytical framework is thus available to find the presence or absence of decay length and oscillatory wavelength in the oscillatory decay of the correlation function. In other words, it became possible to examine the spatial stability of the lane formation more precisely.

*(ii) On the uniformity of lanes in terms of correlation function analysis based on the stochastic DFT*—We have obtained the Fourier transform of the stationary charge–charge correlation function $C_{qq}^{\mathrm{st}}\left(k_{x},k_{y}\right)$. Nevertheless, the previous treatments [9,10,17,18] have supposed that $k_{x}=0$ in advance prior to the inverse Fourier transform. The pole equation obtained from the correlation function at $k_{x}=0$ is an equation in which only $k_{y}$ is a variable, and it is equivalent to the linear stability condition determined from the dispersion relation, as described in the first purpose. Namely, in the previous treatments [9,10,17,18] described above, the presence or absence of lane formation is examined on the premise that the lane formation is uniform in the electric field direction. It is necessary to show the uniformity in the electric field direction itself without assuming the uniformity in the electric field direction. Therefore, we evaluated the Fourier transform of $k_{x}$ from the pole equation for point charge systems (i.e., σ=0) where we have $c^{S}\left(r\right)=0$ and that ω(k)=1. In other words, we investigated the stability of lane formation with only Coulomb interaction at σ=0, showing that the oscillatory wavelength $\lambda_{x}^{*}$ diverges at $k_{y}=k_{y}^{*}$, or the solution to the pole equation given by Equations (Equation 32) or (Equation 37). Thus, the approximation has been validated analytically. Figure 4 also demonstrates numerically that, above the Kirkwood crossover [39,40,41,42], the oscillatory decay length observed for the 3D primitive model (i.e., σ≠0) is longer with the increase of ion density $\bar{n}$; the underscreening behavior under external field applied remains to be investigated in more detail (see also [22]).

## Figures and Tables

**Figure 1 entropy-24-00500-f001:**
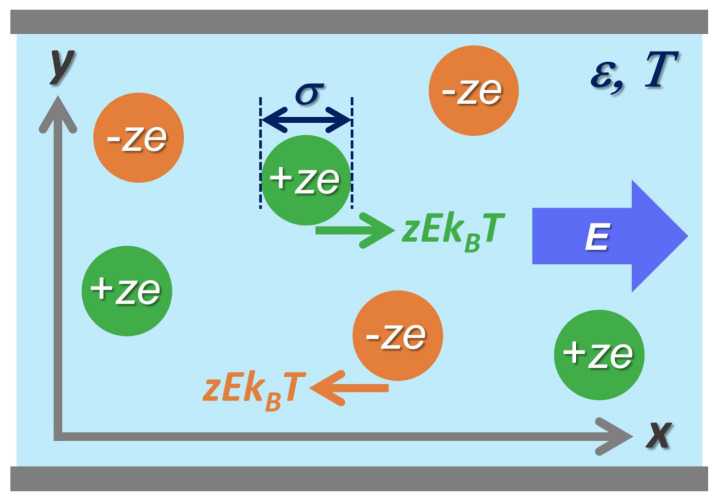
A schematic of the 2D primitive model of binary ionic mixture with a static electric field E applied in the *x*-direction. The *z*-valent cations and anions are modeled by equisized charged hard spheres of diameter σ immersed in a dielectric medium with dielectric constant ϵ at a temperature *T*.

**Figure 2 entropy-24-00500-f002:**
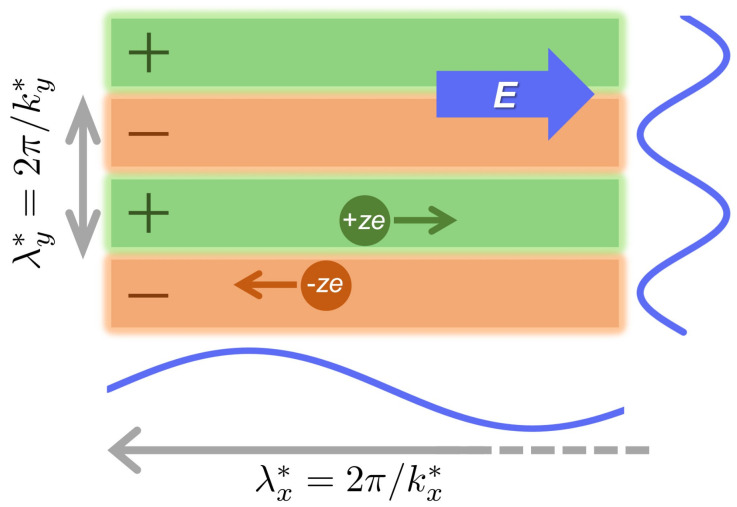
A schematic of lane formation in a binary ionic mixture. The green and orange lanes represent aligned segregation bands of cations and anions, respectively. Correspondingly, the positive and negative signs seen on the lanes indicate that each lane is a mesoscopically charged object. The wavelengths, $\lambda_{x}^{*}$ and $\lambda_{y}^{*}$, in *x*-and *y*-directions are related to wavenumbers as $\lambda_{x}^{*}=2\pi /k_{x}^{*}$ and $\lambda_{y}^{*}=2\pi /k_{y}^{*}$ (i.e., Equation (Equation 58)), respectively. In this study, these wavenumbers are determined by Equations (Equation 32) and (Equation 35) when considering point charges.

**Figure 3 entropy-24-00500-f003:**
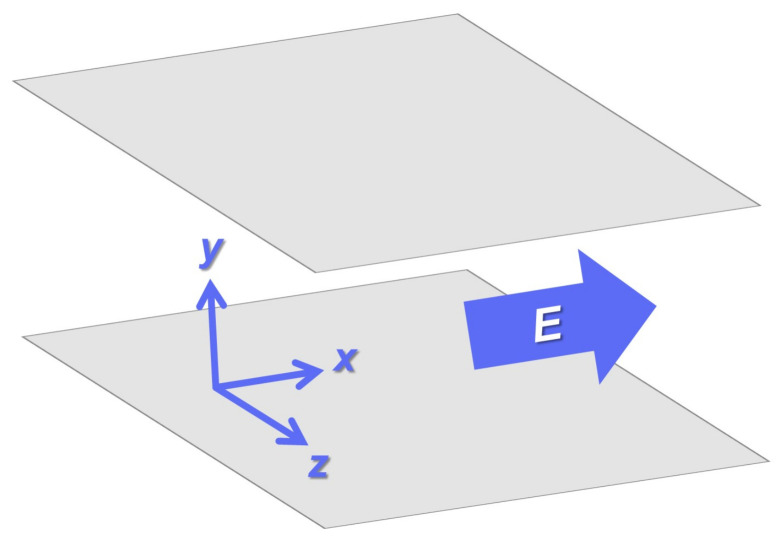
A schematic of the 3D primitive model in Cartesian coordinates illustrates a binary ionic mixture confined between two parallel plates. While the *y*-axis is perpendicular to these plates, the electric field is applied in the *x*-axis.

**Figure 4 entropy-24-00500-f004:**
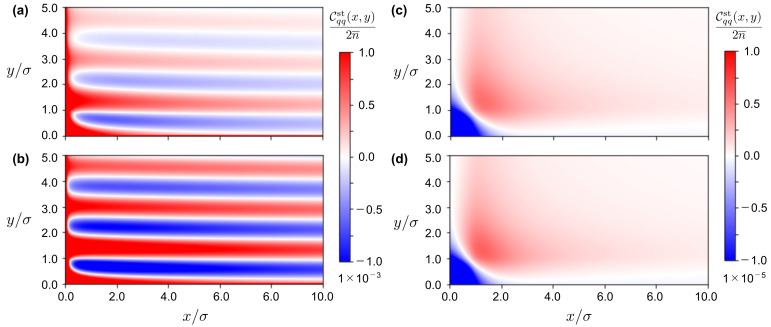
The real-space representation $C_{qq}^{\mathrm{st}}\left(x,y\right)$ of the charge-charge correlation function at zEσ=1.0 is shown using heat maps where the length scale is in units of diameter σ. We obtain the real-space correlation function from performing the 2D inverse Fourier transform of $C_{qq}^{\mathrm{st}}\left(k\right)/\left(2\bar{n}\right)$ given by Equations (Equation 22) and (Equation 23). The remaining parameter set of $\left(\tilde{E},\;\bar{\kappa }\sigma ,\;\phi \right)$ is (**a**) (0.484,1.55,0.05), (**b**) (0.484,1.62,0.055), (**c**) (0.0,1.55,0.05), and (**d**) (0.0,1.62,0.055).

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
