# Peer review of "Stochastic Density Functional Theory on Lane Formation in Electric-Field-Driven Ionic Mixtures: Flow-Kernel-Based Formulation"

_entropy, 2022, doi:10.3390/e24040500_

Round 1

Reviewer 1 Report

In this manuscript the author have theoretically studied the electric-field-driven lane formation in ionic mixtures. While the mathematical framework appears to be correct, I think, in its present form, the manuscript is not suitable for publication. The author may consider the following remarks while revising the draft. 

[1] The introduction section states that the study can be applied to electrolytes with ions, colloidal particles, and ionic liquids. I am not sure how. An ion and a colloid moves in a very different way under the application of an electric field. The author uses the expression electrophoresis several times. While a colloidal particles do undergo electrophoretic motion when applying an electric field, a small ion does not. This should be clarified. 

[2] The paper lacks any insight into the underlying physical mechanism. For example, why do the lanes even form under the application of an electric field? Some abstract equations are not very helpful to grasp the content so easily, particularly for the often not too specialised readers of a journal like Entropy.  

[3] The author should provide a feel for the actual numbers. For example, what is a typical width of a lane? What is the size of the typical particles being considered? How strong an electric field is necessary to apply. What should be the typical charge of the particles?

[4] In Fig. 1 it is perhaps better to sketch a few more particles. 

Author Response

Dear Reviewer,
I am submitting herewith a new manuscript that fully addresses the concerns raised by the reviewers of the above manuscript. We have now added numerical results, in order to overcome the suggested lack of concrete results. I reply to the reviewers' comments with the summary of revisions in the attached file.

I would appreciate it if you could consider the present form, which is revised according to the previous reviewers' comments, for publication in Entropy.

Yours sincerely,
Hiroshi Frusawa
Kochi University of Technology

Reviewer 2 Report

The manuscript entitled <Correlation function analysis of lane formation in electric-field-driven ionic mixtures> is a valid research work with appropriate level of novelty and originality. The topic of the manuscript fits well the scope of the journal. Overall, the topic as actual and very highlighted and manuscript can be of definite interest for the readers, especially those working in the specific field of research.

The introduction section clearly represents the state of the art in the specific field; the section is appropriately supported by relevant references including the recent ones. The manuscript has a very well-organized structure. The obtained results are described and treated correctly. 

With pleasure I can recommend this manuscript for considering it for publication at Energies-MDPI. However, authors should make a minor/technical revision before:

(A) The manuscript should be placed in the specific template in a full accordance with the journal rules. The style of the references should be also in accordance with journal rules.

(B) The style of the manuscript language might be checked by the native speaker; this will increase the overall clarity of the manuscript.

With pleasure I can recommend this manuscript for publication after a minor/technical revision.

Author Response

(The authors gave the same response as above.)

Round 2

Reviewer 1 Report

I find the author's reply to my comments [1], [2], and [4] satisfactory. In particular, the removal of a lot of discussion on colloids and of 'electrophoresis' has improved the clarity as it was misleading. However, I am still not satisfied with the response to my comment [3]. While the benefit of using dimensionless quantities is undeniable, some effort from the author to resume the results in a way that could result in more immediate comprehension of a wider audience is desirable. This would not only help to spread the enthusiasm among the wide authorship of Entropy but will also improve the diffusion of the paper among the experimentalists in the field. Therefore, I ask the author to reconsider comment [3] from the previous report. On a similar note, can the charge-charge correlation function be measured experimentally? Is it possible to at least add some references? In addition, Fig. 4 should contain labels for the axes and for the color bar.
